# Modeling the Properties of Curcumin Derivatives in Relation to the Architecture of the Siloxane Host Matrices

**DOI:** 10.3390/ma15010267

**Published:** 2021-12-30

**Authors:** Florentina Monica Raduly, Valentin Rădiţoiu, Alina Rădiţoiu, Adriana Nicoleta Frone, Cristian Andi Nicolae, Violeta Purcar, Georgiana Ispas, Mariana Constantin, Iuliana Răut

**Affiliations:** 1Laboratory of Functional Dyes and Related Materials, National Research and Development Institute for Chemistry and Petrochemistry—ICECHIM, 202 Splaiul Independentei, 6th District, 060021 Bucharest, Romania; monica.raduly@icechim.ro (F.M.R.); vraditoiu@icechim.ro (V.R.); ciucu_adriana@yahoo.com (A.N.F.); ca_nicolae@yahoo.com (C.A.N.); violeta.purcar@icechim.ro (V.P.); georgiana.ispas23@yahoo.com (G.I.); mariana.calin@icechim.ro (M.C.); iulia_rt@yahoo.com (I.R.); 2Faculty of Pharmacy, Titu Maiorescu University, Bd. Gh. Sincai, No.16, 040441 Bucharest, Romania

**Keywords:** curcumin derivatives, hybrid films, fluorescence emission, antimicrobial activity

## Abstract

Research in the field of natural dyes has constantly focused on methods of conditioning curcumin and diversifying their fields of use. In this study, hybrid materials were obtained from modified silica structures, as host matrices, in which curcumin dyes were embedded. The influence of the silica network structure on the optical properties and the antimicrobial activity of the hybrid materials was monitored. By modifying the ratio between phenyltriethoxysilane:diphenyldimethoxysilane (PTES:DPDMES), it was possible to evaluate the influence the organosilane network modifiers had on the morphostructural characteristics of nanocomposites. The nanosols were obtained by the sol–gel method, in acid catalysis. The nanocomposites obtained were deposited as films on a glass support and showed a transmittance value (T measured at 550 nm) of around 90% and reflectance of about 11%, comparable to the properties of the uncovered support. For the coatings deposited on PET (polyethylene terephthalate) films, these properties remained at average values of T550 = 85% and R550 = 11% without significantly modifying the optical properties of the support. The sequestration of the dye in silica networks reduced the antimicrobial activity of the nanocomposites obtained, by comparison to native dyes. Tests performed on *Candida albicans* fungi showed good results for the two curcumin derivatives embedded in silica networks (11–18 mm) by using the spot inoculation method; in comparison, the alcoholic dye solution has a spot diameter of 20–23 mm. In addition, hybrids with the CA derivative were the most effective (halo diameter of 17–18 mm) in inhibiting the growth of Gram-positive bacteria, compared to the curcumin derivative in alcoholic solution (halo diameter of 21 mm). The results of the study showed that the presence of 20–40% by weight DPDMES in the composition of nanosols is the optimal range for obtaining hybrid films that host curcumin derivatives, with potential uses in the field of optical films or bioactive coatings.

## 1. Introduction

Curcumin is the main compound extracted from *Curcuma longa*. Among the classic uses of curcumin, we can mention its frequent presence in our diet, as a spice or food preservative, and it is often used in Ayurvedic medicine or traditional Chinese medicine in the treatment of various diseases. In modern applications, it is found as food supplements, food additives or cosmetics, adjuvants in cancer or Alzheimer’s treatments. These uses are possible due to its bioactivity demonstrated as antioxidant, antiviral, antibacterial, and antifungal [1,2,3]. Depending on the field of use, curcumin studies have been directed on methods to increase solubility for biomedical applications [4,5,6,7,8] or aimed at improving light or temperature stability properties for unconventional applications, such as sensors [9,10,11], pH indicators [12,13], dyes for solar cells [14,15,16,17], (bio) active coatings [18,19,20,21,22,23,24,25,26]. In this regard, the dye was used in its natural form, or it was chemically modified either on β-diketone or peripheral aromatic groups. The physical and chemical properties of these structures are influenced by the type of dye conditioning, such as micelles [8,27], nanoparticles [6], or composite materials [28,29,30]. The targeted field of application is very important in choosing the type of conditioning of curcumin derivatives [31]. From this point of view, curcumin is used as a food additive in the form of micelles or nanoparticles [32,33,34]. Porous materials and metallic nanoparticles are used as delivery platforms for curcumin and its derivatives which can be used as adjuvants in the treatment of many diseases [35,36,37,38]. Moreover, the antimicrobial properties of curcumin have led to the development of composite materials compatible with the molecular structure of the dye used in order to obtain coatings with antibiofilm properties [39,40]. For this purpose, several types of materials were made and in order to do that curcumin was encapsulated in various polymeric matrices, such as cellulose [41,42], polyvinyl derivatives [43], or siloxanes [44,45]. Each of them has specific characteristics depending on the purpose of use: food packaging [46,47,48,49], coatings of equipment in the food industry, medical devices, surfaces, and utensils for medical use [50]. At the same time, one of the most common methods of obtaining film-forming materials is the sol–gel process. By using this method, a series of nanocomposites based on the silica network generated by triethoxysilane modified with different types of organic silane derivatives with alkyl or aromatic groups can be made. They have the role of creating certain morphological structures in the network which can modify the general properties of nanocomposites and/or can create environments compatible with other organic or inorganic structures, that are to be incorporated in the siloxane matrix. Thus, organic–inorganic hybrid materials are obtained and deposited as thin films [51,52,53]. Depending on the field in which they are used, the nanocomposites must have certain properties and be compatible with the support on which they are deposited [54,55]. In other cases, anticorrosive coatings must have barrier properties and usually contain metal oxides; photoactive coatings used in solar applications must have good anti-reflection and transmitting properties and usually contain dyes, or metal complexes, while anti-biofilm coatings frequently host metallic nanoparticles, natural compounds, or antibiotics and have antimicrobial activity [56,57,58,59,60,61,62].

From this perspective, we conducted a study on the influence of siloxane host matrices on the properties of curcumin dyes, embedded in the silica network. Two curcumin derivatives obtained by synthesis were selected for the study. The first dye (CC) has the same structure as the natural compound, curcumin, and the second (CA) has grafted an acetamido group in the 4th position of both phenyl rings. After modifying the structure of the silica host network with organic residues, the variation of the optical properties, the thermostability, and the antibacterial activity of the dyes embedded in the hybrid network compared to the native dye were evaluated. The study aims to contribute to the development of a pattern in modeling properties of curcumin derivatives, due to the fact that there are different types of interactions established between dyes and hybrid silica hosting matrices.

## 2. Materials and Methods

### 2.1. Materials

#### 2.1.1. Curcumin Dyes

Curcuminoid-like derivatives were synthesized with microwaves and purified using a method already published by our group. A mixture of boron trioxide (4 mmol), acetylacetone (8 mmol), and tributyl borate (3.2 mmol) was introduced in a porcelain capsule and was irradiated in a microwave oven at 300 W for 10 min. After the formation of the boron complex of acetylacetone, aromatic aldehyde (7 mmol), and dodecylamine (0.162 mmol) were added, and then the mixture was irradiated for another 20 min at 100–500 W (depending on the aromatic aldehyde chemical structure). An aqueous solution of acetic acid (10% by weight) was added to the reaction mixture, and the obtained suspension was filtered off and the solid product was washed with cold water then dried. The obtained product was purified by recrystallization from a mixture of ethyl-acetate: methanol = 3:2 (*v*/*v*) [63].

#### 2.1.2. The Host Matrix

The following raw materials were used to make the siloxane host matrices: tetraethylortosilicate (98%, TEOS, Aldrich, St. Louis, MO, USA), phenyltriethoxysilane (97%, PTES, Aldrich, St. Louis, MO, USA), diphenyldimethoxysilane (98%, DPDMES, Aldrich, St. Louis, MO, USA), ethanol (96%, EtOH, Chimreactiv, Bucharest, Romania), tetrahydrofuran (99%, THF, Merck, Kenilworth, NJ, USA), hydrochloric acid (0.1 N, HCl, Chimreactiv, Bucharest, Romania). The obtained films were deposited on biaxially-oriented polyethylene terephthalate films (PET, Tg = 105 °C, 0.075 mm thick, Xerox Co., Norwalk, CT, USA) and microscope slides (ISOLAB, 1 mm thick, purchased from AMEX, Bucharest, Romania).

#### 2.1.3. Microorganisms

For the evaluation of antibacterial activity, the following strains of bacteria were used: *Staphylococcus aureus*, ATCC 25,923 (*S. aureus*), *Escherichia coli*, ATCC 25,922 (*E. coli*), and a strain of fungi: *Candida albicans*, ATCC 10231(*C. albicans*) from Microbial Collection of ICECHIM.

### 2.2. Methods

#### 2.2.1. Obtaining the Mixture of Film-Forming Materials

The siloxane polymer matrix was obtained by using the sol–gel process in acid catalysis [53]. By varying the silane precursors TEOS, PTES, DPDMES, and the mass ratio between them, four types of polymeric materials were obtained, in which curcumin derivatives were embedded (Table 1). After mixing the raw materials for 5 h at room temperature, homogeneous solutions were obtained which were deposited by dip-coating on two types of substrate (Figure 1). The first type of supports used were plastic slides made of transparent polyethylenterephthalate (PET), while the other type of substrates used were microscope glass slides which, before use, were cleaned with a 30% hydrogen peroxide solution, then with acetone and after that dried. After the deposition process, the films were dried at room temperature for 24 h, then put in the oven at 110 °C for 2 h.

#### 2.2.2. Characterization Methods

The films obtained were mapped using a microscope produced by VICKERS-JOYCE LOEBL and equipped with an OPTIKA 3 Mpx optical camera, and the images were processed with ProView software.

The structural differences of the host matrices were analyzed by FTIR measurements, recorded with a JASCO FT-IR 6300 instrument (Jasco Int. Co. Ltd., Tokyo, Japan), equipped with a Specac ATR Golden Gate (Specac Ltd., Orpington, UK) with KRS5 lens, in the range of 400 to 4000 cm^−1^ (32 accumulations at a resolution of 4 cm^−1^).

Thermogravimetric analysis of the hybrid materials obtained was performed with a TGA Q5000IR instrument (TA Instruments, New Castle, DE, USA). The 10–15 mg samples were analyzed in platinum pans under the following conditions: ramp 10 °C/min to 700 °C, isothermal for 5 min, two gases (purge gas 1: Nitrogen 5.0 (99.999%), 50 mL/min and purge gas 2: Synthetic Air 5.0 (99.999%), 50 mL/min).

The surface morphology of film surfaces deposited on PET lamella was determined using a MultiMode 8 atomic force microscope (AFM) (Bruker, Santa Barbara, CA, USA). Measurements were carried out in Peak Force (PF) Quantitative Nanomechanical Mapping (QNM) mode, in air, at room temperature using silicon nitride tips at a scan rate of 1 Hz and a scan angle of 90°. The data and images were processed with the NanoScope software version 1.20. The root-mean-square roughness (Rq) was used to express the surface roughness of different materials and was calculated using the NanoScope AFM software. At least four unprocessed AFM topographic images of 5 × 5 μm^2^ were analyzed for each sample.

Contact angles of films were determined with a CAM 200 (KSV Instruments, Helsinki, Finland) equipped with a high-resolution camera (Basler A602f, Basler, Ahrensburg, Germany) and an auto-dispenser. The contact angles were measured in air, at room temperature, and at ambient humidity 2 s after the drop contacted the surface of the films. Drops of 6 μL deionized water were dispensed on each sample and the value of the reported contact angles was the average of six measurements. The data was processed with CAM software.

The fluorescence properties of the coatings deposited on the two substrate types were analyzed by fluorescence spectra recorded on a JASCO FP 6500 spectrofluorometer (Jasco Int. Co. Ltd., Tokyo, Japan), at 25 °C.

The diffuse reflectance and transmittance spectra of the hybrid films obtained were recorded on a JASCO V570 UV-VIS-NIR spectrophotometer (Jasco Int. Co. Ltd., Tokyo, Japan), equipped with a JASCO ILN-472 (150 mm) integrating sphere, using uncovered support as reference.

Tests for the antimicrobial activity of nanocomposites were performed in Petri dishes on a specific agarized medium: Sabouraud medium for *C. albicans* and Muller Hinton medium for *E. coli* and *S. aureus* bacteria. The evaluation of the antimicrobial activity was performed by using the diffusimetric disc method, with inoculation in discs and spots (10 µL), on Petri dishes, in an agarized medium inoculated in the canvas with the tested microorganisms. The plates were incubated for 24 h at 28 °C for *C. albicans* and at 37 °C for bacteria. The antimicrobial activity was evaluated by measuring the diameter of the clear area (halo) that appears around the inoculation point (spot or paper disc). The samples were performed in duplicate.

## 3. Results

Several different types of nanocomposites were obtained by modifying the mass ratio between the TEOS network generator and the PTES and DPDMES network modifiers (Table 1). Nanosols were deposited as films which, as reported in the literature, [53,54,55] have different properties due to the intermolecular bonds established with the support and influenced by the hydrophobicity of the surfaces on which they are deposited (Figure 2).

### 3.1. Morphological Characterization of Host Matrices and Composite Materials

#### 3.1.1. Characterization of Host Matrices by ATR-FTIR Spectra

Siloxane-type host matrices were characterized by the FTIR method (Figure 3) to further demonstrate the relationship between the compositions of nanosols and the properties of the hybrid materials obtained. Given that the composition of the silane matrices differs by the mass ratio between the silane network modifiers and the presence or absence of the TEOS network generator, the FTIR spectra were similar. Thus, bands characteristic to the hydrogen bonds between the O-H groups were present at 3334–3351 cm^−1^. The signals from 3051–3071 cm^−1^ were attributed to the stretching vibrations of the CH aromatic groups. Bands at 1592–1595 cm^−1^ and 1429–1431 cm^−1^ were characteristic to phenyl groups, with higher intensity in the case of M3 and M4 compounds in which organosilane compounds were in a higher proportion. For these compounds, for the same reason, the intensification of the signals from 735, 718, and 693 cm^−1^ characteristics to the C-H group from the monosubstituted aromatic groups were also observed. For the matrices, M1 and M2 a wide band was observed at 1020, 1013 cm^−1^, respectively, which are characteristic of the Si-O-Si (stretching vibrations), and the presence of a weak signal at 1131 cm^−1^ attributed to the Si-O-C stretching vibration was also observed. As the proportion of organosilane compounds increased, there was an intensification of the signal’s characteristic to Si-O-C and a displacement of the peak to 1128 cm^−1^. At the same time, the band from 1020 cm^−1^ corresponding to the Si-O-Si was affected by the significant presence of aromatic groups and increased, highlighting three maximum values at 1050, 1013, and 993 cm^−1^.

Because the content of the dyes in the film-forming material was very small, their structural features were more difficult to detect by using FTIR spectrometry. However, in the case of CA, there were two characteristic functional groups that had very high intensity, namely the acetamide group which had the band characteristic of the carbonyl group located at 1662 cm^−1^, and the band characteristic of the keto-enol equilibrium of the acetylacetone rest, located at 1633 cm^−1^. These were found in the FTIR spectrum of M1-CA, embedded in the wide and asymmetric band, characteristic to water bending vibration, which had a maximum at 1623 cm^−1^, as can be seen from the zoom zone in Figure 3b.

#### 3.1.2. Thermogravimetric Analysis of Composite Materials

From the thermogravimetric analysis of the hybrid materials, it was observed that up to 120 °C, the matrices of type M1-CC (Figure 4a), M1-CA, respectively (Figure 4b) with high content of TEOS, had a high weight loss of about 3% compared to those in which the amount of TEOS was reduced (M2-CC and M3-CC) or missing (M4-CC).

This mass loss is attributed to the water and solvents used to obtain the nanosols. In the range of 120 to 350 °C, the mass losses are attributed to the interstitial water. These losses were lower for M1 type materials and increased for the other materials between 5.9–8.19%, with an increase in the cavities in the silica network, determined by the presence of organosilane network modifiers. The third stage of decomposition took place in the range of 350 to 540 °C, where organic compounds were lost (aromatic groups from the silica network and dye). In this case, the hybrid material M4-CC had the biggest loss, of approximately 63.76%, followed by M2-CC with a loss of 35.55%. These two materials had the highest ratio of DPDMES in their composition. In the last range of temperature 540–700 °C, the silica network was degraded by breaking the weak Si-OR and decomposing organic residues with losses between 6.28% for M4-CC and 15.79% for M3-CC.

Following the results obtained from the thermogravimetric analysis (Table 2) it can be observed that up to 350 °C, all composite materials lose approximately 6–7% of the initial mass. This process took place in two stages with approximately equal percentages for M1-type materials with high TEOS content, without being influenced by the type of encapsulated dye (Figure 4b). This decomposition process was directly related to the degradation of the siloxane matrix, the pure dyes showing decomposition steps at lower temperatures (Figure 4c). As shown in Table 2, the maximum mass loss occurred at 262 °C for curcumin and 296 °C for the CA derivative. The role of TEOS in the formation of the silica network was obvious in the case of M2-M4 materials in which its quantity was small or missing. For these materials, the mass losses were below 0.5% in the first stage, and in the second stage, the temperature of maximum decomposition rate increased when a much higher mass loss occurs. The M2-CC and M3-CC composite materials differed from each other by the ratio between PTES: DPDMES. In the case of M2-CC, the ratio was 1.25:1, the residue at 700 °C in air was 26.38% which represented inorganic silica residues, while in M3-CC the ratio was 3.5:1 residue at 700 °C, in air was 40.06%. However, the highest mass loss occurred in the case of M4-CC nanocomposites, whose composition lacked the network generator. The absence of TEOS in the nanosol affected the architecture of the film due to the lack of highly cross-linked organized structures that were generated. For this reason, the intermolecular forces of π–π or van der Waals type, present in M4-CC, were easier to break and led to higher mass loss. The absence of TEOS was also reflected in the reduced amount of residue, respectively, 12% compared to the initial mass. Thus, we can conclude that PTES, due to conformation, can be integrated into the structure of the silica network, while for DPDMES, due to steric effects, phenyl groups are more labile.

#### 3.1.3. Determination of the Morphological Properties of Nanocoatings by AFM

The topographical analysis of the films obtained in the case of M4, in which the crosslinking agent (TEOS) was missing, showed that the relatively high proportion of DPDMES led to the formation of linear structures suitable for obtaining film-forming materials, but containing agglomerated formations caused by PTES which formed silica networks around associates of π–π stacking type due to the presence of benzene nuclei in both components, having R_max_ = 500–600 nm. In the case of M4-CC, the introduction of the dye led to obtaining less prominent structures, having R_max_ = 200–250 nm, which were better dispersed in the film-forming material. The effect of adding the dye appeared to make the components more compatible by van der Waals forces between the benzene nuclei present in all three components of the filmogen. This behavior also takes into account the flatness of the dye molecules together with the linear structures generated by DPDMES. The effect generated in this case created the appearance of “orange peel”, at the nanometric level (Figure 5b).

The influence of the ratio between the modifying components of the silica film-forming network, generated from TEOS in the presence of the dye, can be seen very well from Figure 5c,d. In the first case, the existence of a higher amount of DPDMES in the film-forming composition, to the detriment of PTES, led to the formation of films with lower root-mean-square roughness Rq = 1.51 nm (standard deviation 0.43 nm), while, in the case of M3-CC, the obtained films had a root-mean-square roughness of Rq = 6.41 nm (standard deviation 0.65 nm). Although it had a lower roughness, in the case of M2-CC, the existence of formations with diameters larger than 1.6 µm was observed, probably caused by the segregation of a part of linear siloxane generated by the higher amount of DPDMES, while, in the case of M3-CC, segregations were much rarer and they had diameters below 1.1 µm.

#### 3.1.4. Contact Angle Measurements

From literature, it is known [56,57,58] that the surface of the untreated glass has a hydrophilic character due to the existence of Si-OH groups at the surface while the PET surface manifests a less hydrophilic character due to carbonyl and phenyl groups. The nanosols obtained, depending on the ratio between the network initiator (TEOS) and the organosilane network modifiers, could be tuned for the advancing or receding contact angles of water at the surface of the hybrid films generated through sol–gel processes [59]. This had an effect on the compatibility of hybrid materials with the substrate; the glass coatings considerably improved anti-welding properties, with a contact angle of around 79 ± 2°, exceeding that of glass support (Figure 6). The less hydrophilic character of nanosols given by the network modifiers and by the dye embedded in the silica network competed with the properties of PET without significantly modifying the contact angle (77 ± 2°) of the support.

The orientation at the interface with the support of the hybrid films was mainly towards the glass for the silanol groups and towards the air for the phenyl groups, which made the differences of contact angle with the support bigger in the case of the glass. Regarding the PET support at the interface with the film former, interactions occurred through hydrogen bonds between the silanol groups in the film and the ester groups in the support, as well as through π–π interactions between the phenyl groups in the film and those in the support. Thus, the very small difference in water contact angle between the coated material and the PET support is explained.

### 3.2. Photophysical Properties of Nanocomposite Coatings

#### 3.2.1. Fluorescent Properties of Nanocomposites Deposited on Glass/PET Supports

The nanocomposites obtained were deposited on two types of support, PET (Figure 7a,b) and glass support (Figure 7c,d). The films obtained on PET showed three peaks with different intensities for nanocomposites containing curcumin (Figure 7a) and two peaks at different intensities for nanocomposites containing the curcumin derivative (Figure 7b). The maximum of fluorescence intensity in the case of the dye with curcumin structure (CC) was situated at 525 nm and 510 nm for CA. These maxima were given by the presence of the dye in the matrix, with a bathofluoric shift due to intermolecular interactions by hydrogen bonds. Pure curcumin derivatives in the polar environment of the silica matrix were characterized by the fluorescent emission peak at 545 nm for curcumin and 530 nm for the CA derivative [63]. The peak at 420 nm was given by the intermolecular interactions between the dye and the host matrix, as already presented by the authors in another paper [53]. The keto-enol and cis-trans tautomeric structures of the dyes were influenced by the size of the cages formed in the silica network, by the organic network modifiers PTES, DPDMES, and by the traces of polar solvents left in the cavities [4], following the sol–gel process. Moreover, it can be observed that the presence of different auxochromes on the aromatic groups influenced both the interactions with the host matrix and with the support on which the films were deposited. Thus, the curcumin-containing films deposited on PET, compared to those deposited on glass slides (Figure 7c), showed a shoulder at 480 nm, due to the π–π type interactions with the carbonyl groups of the support. In the case of nanocomposites deposited on the glass support, the interactions of the films with the support took place through intermolecular hydrogen bonds that led to hypsochromic displacement and a dramatic decrease in fluorescence intensity. The same phenomena occurred with a decrease in the dye loadings in the silica matrix (Figure 7d).

#### 3.2.2. Spectrophotometric Properties of Nanocomposites Deposited on Glass/PET Support

For hybrids of M2 type, UV-Vis measurements were performed in order to evaluate the optical characteristics regarding the reflectance (Figure 8) and transmittance (Figure 9) of the obtained films. It can be seen that the properties of the coatings in which the two chromophores were present were similar along the visible spectrum, in the absorption range of pure dyes at 424 nm for CC and 419 nm for the CA derivative [63]. The differences between the two types of M2-CC and M2-CA coatings were mainly due to the intermolecular interactions established between the nanocomposites and the support. The optical properties were compared by measuring the UV-Vis spectra of reflectance and transmittance at 550 nm, and the values for the investigated samples are reported in Table 3.

From the data presented in Table 3, it can be observed that the silane matrix increased the reflectance of the glass support, the coatings being compatible with the hydrophilic surface of the glass through intermolecular hydrogen bonds. The films deposited on PET led to a decrease in reflectance, probably due to the segregation processes that appeared in the silica coatings [51,62]. At the same time, it can be observed that the presence of the curcumin dye in the polymer matrix did not significantly influence the reflection properties of the coatings, regardless of the concentration or the type of the coated support. The role of the dye resulted from the transmittance values measured at 550 nm. Thus, it can be seen that coating the glass support with silica nanocomposites reduced the transmission properties of the glass. These properties were enhanced by the addition of the curcumin dye through its fluorescence emission processes in the visible range and the intermolecular bonds established with the host matrix. Based on these considerations, it can be seen that a decrease in the dye concentration to 25% had a significant effect on increasing the transmittance of the film. The steric effects were evident in the curcumin derivative where the acetamide group, more voluminous than the hydroxyl and methoxy groups in curcumin, prevented the aggregation processes. The transmittance of the films in which the CA derivative was incorporated was very good even at higher concentrations compared to that of hybrid films with curcumin. In the case of the PET support coated with M2-type nanocomposites, the transmittance decreased compared to the uncovered support. The presence of the chromophore in the silane matrix compensated for these shortcomings and managed to improve the transmittance of the coated support at the level of the uncovered one.

#### 3.2.3. Study of the Antimicrobial Activity of Nanocomposites

Curcumin encapsulated in various polymeric systems [64,65,66] was tested and demonstrated antimicrobial activity. The antimicrobial properties of the nanocomposites obtained in this study were tested against *Candida albicans*, *Staphylococcus aureus*, and *Escherichia coli*. The results were compared with the antimicrobial activity of the dye in 0.3% alcoholic solution and with the host matrix (Figure 10a,b). Studies of the antimicrobial activity of nanocomposites were performed by the disc diffusimetric method with disc and spot inoculation (10 µL) (Figure 10c).

Due to the conditioning process of the dye by embedding in silane matrices, it was observed that its antimicrobial activity decreased. This process was determined by the size of the cages made by the network modifiers in nanosols and led to a decrease in the antifungal activity by 30–70% compared to CC, 40–45% compared to native CA, respectively. Antibacterial activity was reduced by 25–60% for nanocomposites containing CC and by 25–50% for nanocomposites with CA, compared to the dye in alcoholic solution (Figure 10a,b). At the same time, the host matrices were tested to demonstrate that the antimicrobial activity of nanocomposites was not only due to the antibiophilic and hydrophobic effect of silane coatings [67].

Spot inoculation had the highest efficiency in antimicrobial activity. Tests performed by the disc inoculation method had poorer results due to the hydrophobic properties of the nanosols which led to slowing the migration processes in the paper support. However, the results demonstrated the presence of antibacterial activity. The nanocomposites tested had the best growth inhibitory activity in the *S. aureus* strain. Out of these, hybrids with the CA derivative were the most effective (17–18 mm) in inhibiting the growth of Gram-positive bacteria, compared to the curcumin derivative in alcoholic solution (21 mm). The results of the tests performed on *E. coli* had modest results of 11–13 mm for the spot inoculation method, respectively, 4–11 mm for the disc inoculation. Tests performed on *C. albicans* fungi showed good results for the two curcumin derivatives embedded in silica networks (11–18 mm) by the spot inoculation method, compared to the spot diameter of 20–23 mm in an alcoholic dye solution. The evaluation regarding the antifungal activity of the nanocomposites was also done by the method of inoculation in the disk, the diameter of the halo being smaller due to the diffusion processes. It can be seen that the antimicrobial activity of nanocomposites depended on the nature of the silica network. For this reason, M4 type hybrids with large pores showed better antimicrobial activity than other types of silane structures, the dye migrating more easily to the pore surface.

## 4. Discussion

Recent research has focused on unconventional applications of curcumin in various fields [31]. Many of the studies considered the conditioning of curcumin in order to successfully capitalize the optical properties and antimicrobial activity of the dye. These bioactive coatings can be applied in the field of biomedical, food, or solar cells. At the same time, the properties of the dye that we intended to use and the influence of the type of conditioning on these properties must be taken into account. The molecular structure of curcumin derivatives has several tautomers due to the double bonds on the hexadiene chain that manifest the cis-trans isomers and the two carbonyl groups that are found in the form of keto-enol [31]. There are also studies on the influence of solvents on the stability of isomers, the enol form being stable in the presence of polar solvents. The influence of auxochromes grafted on marginal aromatic groups of the hexadiene chain also contributes to these factors. Depending on their type, they can establish intra- and intermolecular bonds, such as ionic or covalent bonds [4,5].

For these reasons, two types of curcumin derivatives embedded in siloxane matrices were studied by the sol–gel process. The influence of the silica network structure on the optical properties and antimicrobial activity of dyes was followed. The mass ratio between PTES:DPDMES was modified and the influence of organosilane network modifiers on the morphostructural characteristics of nanocomposites was evaluated. The results of the thermogravimetric analysis showed that the presence of compounds with phenyl groups in the silica network led to a decrease in the thermostability of the network structure (Figure 4). Increasing the number of aromatic groups in the network by adding 20–40% DPDMES reduced the number of Si-O-Si bonds, and led to the formation of large cavities that weakened the structural strength and stability of the silica network.

The hydrophobic character of curcumin derivatives has a significant role in the process of incorporating the dye in the silica network. In order to increase the compatibility of the dye with the inorganic host matrix, the effect of the introduction of phenyl groups in the silica network was studied. The contact angle of the films deposited on the same type of support showed that they were not greatly influenced by the proportion of aromatic groups in the network (Figure 6). These results showed that the phenyl groups were oriented inside the network, influencing the ionic character of the cavities formed, improving the compatibility between the dye and the host matrix. Studies have shown that hybrid films are affected by the support type. The hydrophobic character of PET influenced the ionic character of nanosols, favoring the aggregation processes of dyes (Figure 2). These processes competed with the positive mesomeric effect of the auxochromes grafted on the aromatic groups of the dye that were more pronounced and led to increased fluorescence of the coatings on PET support. Nanocomposites deposited on the glass support had much lower fluorescence intensities. This was due to n-π type intermolecular bonds or hydrogen bonds established by the nanocomposite with the Si-OH groups of the glass. This decrease in fluorescence was also influenced by the stability of the isomer in the enol form of the dye in the presence of the residual polar solvent in the cavities of the silica network. The stability of the cis-trans and keto-enol tautomeric forms of the curcumin derivatives embedded in the host matrix was influenced by the percentage of organosilane network modifiers added. Their ratio determined the size and polarity of the cavities generated in the silica network by the sol–gel process. Depending on the ionic structure of these cavities, the content of THF/EtOH solvents, and the pH of the environment, the structure of the dye underwent several mesomeric transformations when interacting with light radiation, generating several maximums of fluorescence emission at different wavelengths and intensities (Figure 7). Following the performance of the fluorescence spectra, M2 hybrid films were evaluated for spectrophotometric properties. Dye-free M2 coatings reduced the transmittance properties of the substrate due to the thickness of the film. The attempt to correct this shortcoming by incorporating curcumin derivatives through fluorescence processes managed to compensate for the transmission deficit of light radiation, improving the optical performance of coatings as well as those of the uncovered support (Table 3). The incorporation of the dye in the silica networks led, as expected, to a reduction in the antimicrobial activity of the obtained nanocomposites. Although it did not exceed the level of antimicrobial activity of Ciprofloxacin (Figure 10), the existence of antibacterial properties of nanocomposites could be noticed. These properties could be improved by increasing the amount of dye in the nanosols composition. Regarding the antifungal properties, the results of the tests revealed an activity comparable to that of Fluconazole, an antibiotic known in the treatment of fungal infections [68,69]. The conditioning method, through the architecture of the silica network, directly influenced the antibacterial and antifungal properties of the curcumin derivatives.

However, the fact that these properties were not suppressed, opens up new fields of application of hybrid films as bioactive coatings. For these, it is necessary to continue the studies regarding the in vivo/in vitro testing of the obtained nanocomposites. It is further considered to continue the studies regarding the structural modification of the host matrix, in order to improve the optical performances. At the same time, based on the results of this study, the hydrophobic properties of the coatings can be improved by replacing the network modifiers with other organosilane precursors. Depending on the field of application, these types of nanocomposites can be developed with other chromophore structures embedded in siloxane matrices with improved antimicrobial properties.

## 5. Conclusions

In this study, we looked at the influence of the characteristics of the siloxane host matrix on the optical and antimicrobial properties of curcumin derivatives. In this sense, four types of host matrices were obtained by modifying the ratio of alkoxysilanes (PTES:DPDMES) used as network modifiers, in order to accommodate embedded chromophores. The nanocomposites obtained were deposited as films on glass support and showed a transmittance value of around 90% and reflectance of about 11%, similar to the values of the uncovered support. For the coatings deposited on PET films, these properties remained around 85% for T550 and of 11% for R550 without significantly modifying the initial optical properties of the support. Evaluations of the antimicrobial activity of nanocomposites were observed by the method of disc and spot inoculation. The test results revealed the existence of antimicrobial activity of nanocomposites by the formation of a halo with different diameters around the sample that depended on the architecture of the matrix in which the curcumin derivative was incorporated. After the morpho-structural characterization of nanocomposites and the evaluation of optical properties, it was concluded that the presence of 20–40% DPDMES in the composition of nanosols is the range in which it is possible to optimize organic-silica hybrid films hosting curcumin derivatives, useful for applications in the field of solar cells, food packaging or as bioactive coatings.

## Figures and Tables

**Figure 1 materials-15-00267-f001:**
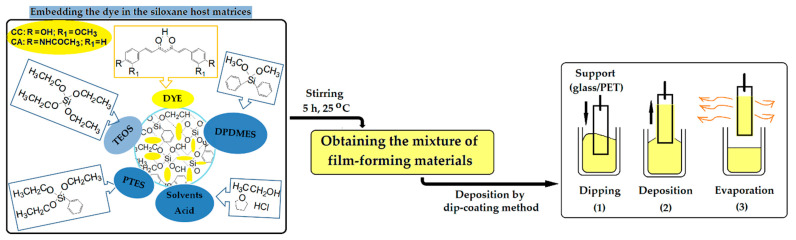
Synthesis of colored nanocomposites, based on siloxane polymer matrix and deposition of nanosols on the support (PET/glass) by dip-coating method.

**Figure 2 materials-15-00267-f002:**
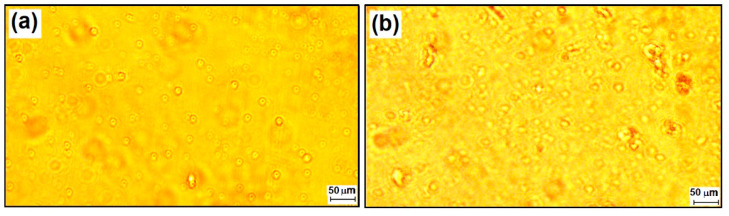
Optical images of M2-CC nanocomposites deposited as films on glass (**a**) and PET supports (**b**).

**Figure 3 materials-15-00267-f003:**
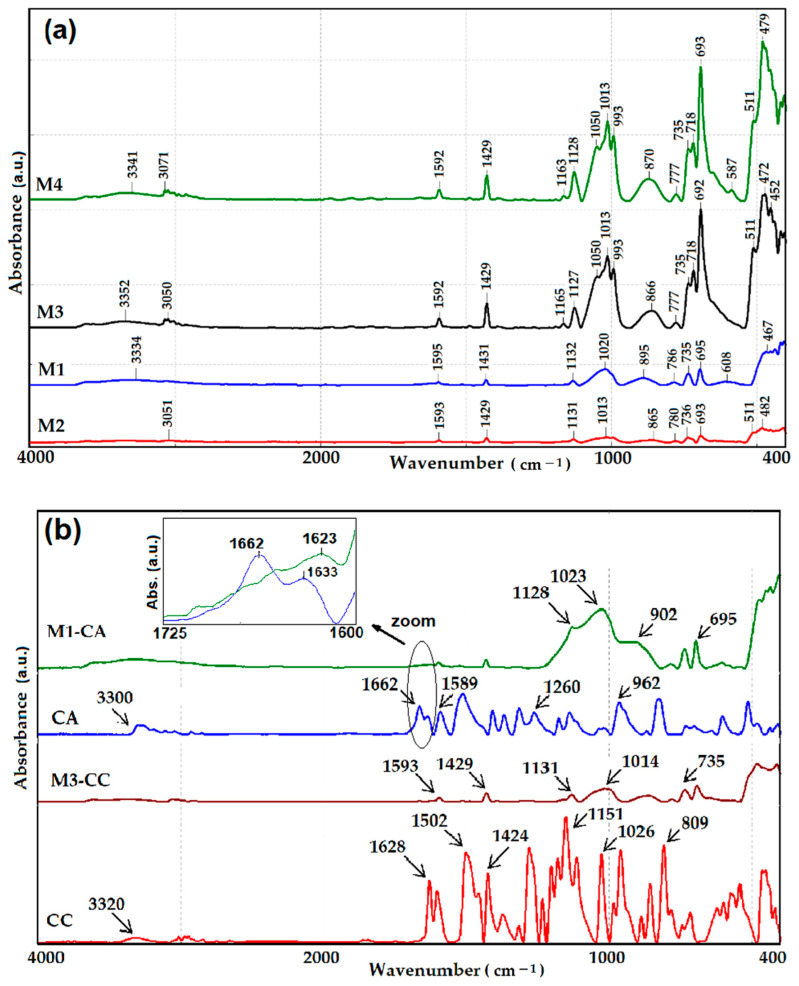
ATR-FTIR spectra of siloxane-type host matrices (**a**) and spectra of pure dyes and dyes embedded in matrices (**b**).

**Figure 4 materials-15-00267-f004:**
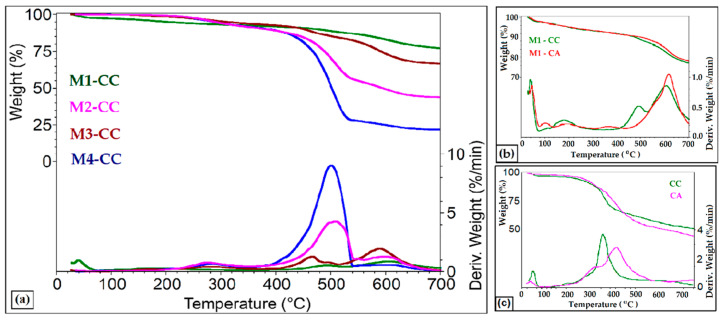
Thermogravimetric curves of hybrid materials, with different siloxane host matrices (**a**) or different dyes (**b**) and pure dyes (**c**).

**Figure 5 materials-15-00267-f005:**
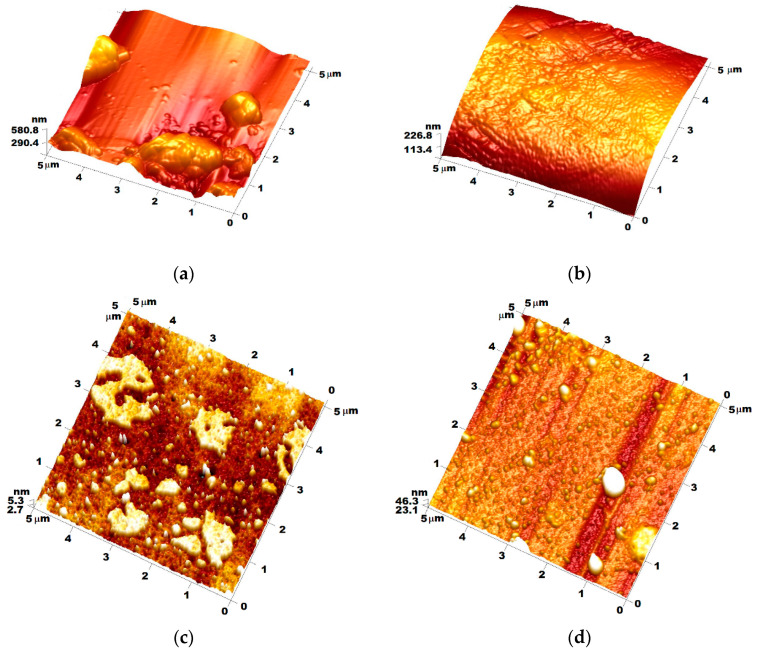
Topography AFM images of (**a**) M4, (**b**) M4-CC, (**c**) M2-CC, (**d**) M3-CC.

**Figure 6 materials-15-00267-f006:**
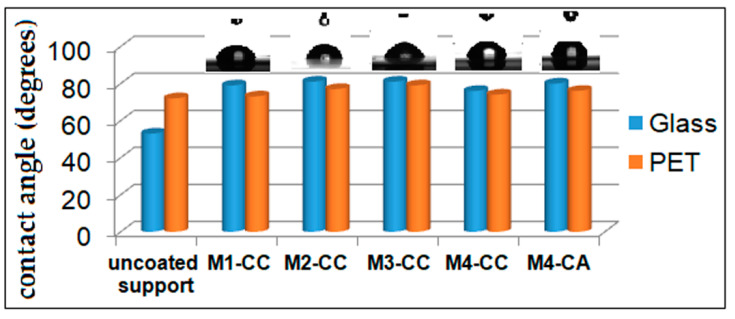
The contact angle of hybrid coatings depends on the coated support.

**Figure 7 materials-15-00267-f007:**
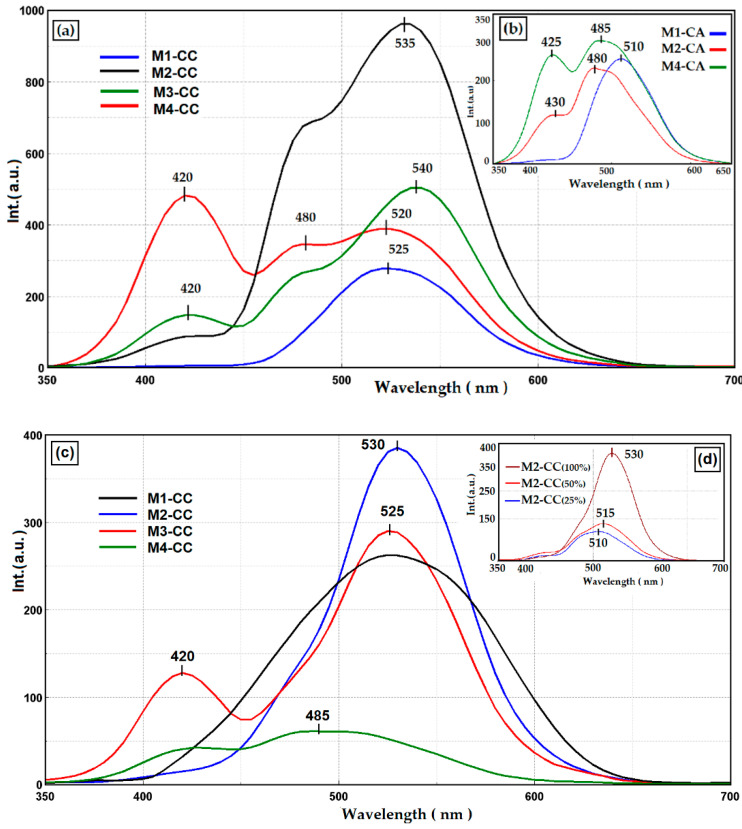
Fluorescence emission spectra of (**a**) hybrid films with curcumin and (**b**) curcumin derivatives deposited on PET, (**c**) hybrids from different matrices with curcumin, and (**d**) with different curcumin loadings, deposited on glass.

**Figure 8 materials-15-00267-f008:**
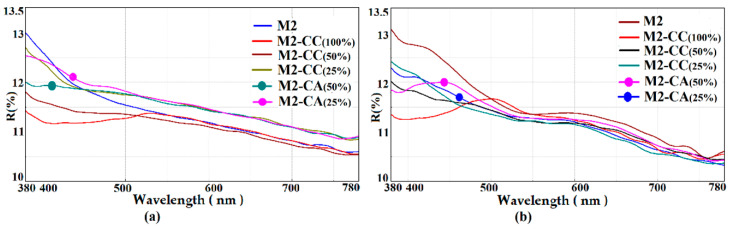
UV-Vis reflectance spectra of M2 hybrid coatings with different dyes loadings deposited on (**a**) glass, (**b**) PET.

**Figure 9 materials-15-00267-f009:**
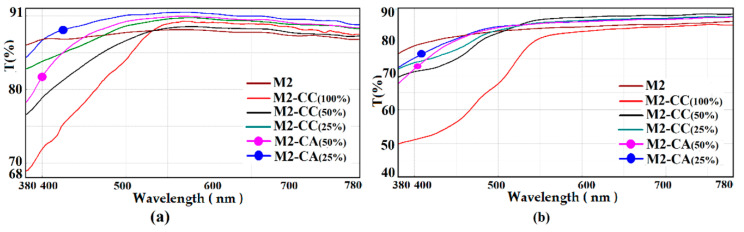
UV-Vis transmission spectra of M2 type hybrid coatings with dyes at variable concentrations deposited on glass (**a**)/PET (**b**) supports.

**Figure 10 materials-15-00267-f010:**
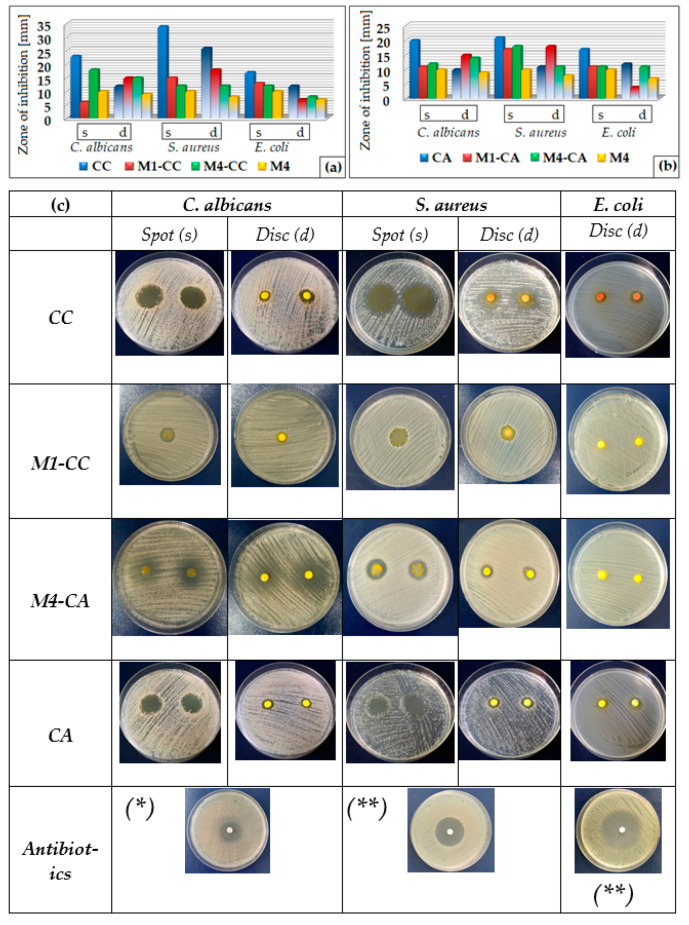
The antimicrobial activity of hybrid materials with (**a**) CC, (**b**) CA, and (**c**) the difference between the antimicrobial activity of the alcoholic solutions of the native dyes, that of the dyes encapsulated in siloxane matrices, and that of the antibiotics (* Fluconazole (25 µg/mL), ** Ciprofloxacin (5 µg/mL)).

**Table 1 materials-15-00267-t001:** The composition of the obtained nanosols.

	Materials	TEOS [mL]	PTES [mL]	DPDMES [mL]	EtOH [mL]	THF [mL]	HCl [mL]	Dye (CC/CA) [g]
Films								
M1		1.5	1.5		2.2	2	0.1	
M1-Dye		1.5	1.5		2.2	2	0.1	0.02
M2		0.3	1.5	1.2	2.2	2	0.1	
M2-Dye		0.3	1.5	1.2	2.2	2	0.1	0.02
M3		0.3	2.1	0.6	2.2	2	0.1	
M3-Dye		0.3	2.1	0.6	2.2	2	0.1	0.02
M4			2.4	0.6	2.2	2	0.1	
M4-Dye			2.4	0.6	2.2	2	0.1	0.02

**Table 2 materials-15-00267-t002:** Characteristic decomposition temperatures and mass losses.

Sample	RT-120 °C	120–350 °C		350–540 °C		540–700 °C		Residue at 700 °C	
	Wt. Loss	Wt. Loss	T_max_	Wt. Loss	T_max_	Wt. Loss	T_max_	N_2_	Air
	(%)	(%)	(°C)	(%)	(°C)	(%)	(°C)	(%)	(%)
M1-CA	3.23	4.14	370.3	3.95		10.53	618.7	78.13	58.32
M1-CC	3.14	4.32	177.0	5.57	490.5	9.89	607.5	77.09	58.79
M2-CC	0.27	8.19	274.4	35.55	509.0	12.37	599.3	43.61	26.38
M3-CC	0.30	5.90	283.0	11.40	466.4	15.79	589.2	66.61	40.06
M4-CC	0.31	7.92	282.9	63.76	500.7	6.28	598.5	21.75	12.30
CC	4.71	7.25	261.6	30.53	327.5	5.91		51.60	12.51
CA	3.26	7.91	296.2	36.73	383.5	7.16		44.94	0.39

**Table 3 materials-15-00267-t003:** UV-Vis reflectance and transmittance at 550 nm for the films with different dyes concentration, deposited on glass/PET support.

Film Sample (Dye Concentration, %)	Glass Support		PET Support	
	R_550_ (%)	T_550_ (%)	R_550_ (%)	T_550_ (%)
Support	10.3	91	12	85
M2	11.4	88	11.4	84
M2-CC (100%)	11.3	89	11.3	81
M2-CC (50%)	11.2	89	11.2	86
M2-CC (25%)	11.2	90	11.2	86
M2-CA (50%)	11.6	90	11.3	85
M2-CA (25%)	11.6	90	11.3	85

## Data Availability

The data are not publicly available due to their containing information that could compromise the privacy of research participants.

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
