# Peer review of "Modeling the Properties of Curcumin Derivatives in Relation to the Architecture of the Siloxane Host Matrices"

_materials, 2021, doi:10.3390/ma15010267_

Round 1

Reviewer 1 Report

This manuscript introduced the “Modeling the properties of curcumin derivatives in relation to the architecture of the siloxane host matrices.”

Some minor revisions are required to enhance quality of research article.

  1. The abstract and conclusion should precise and need to rewrite.
  2. The authors used two curcumin derivatives to embed in siloxane polymer matrix, synthesized in laboratory. To increase more clarity about synthesized material, give comparison of decomposition of pure curcumin derivatives with composite materials.
  3. Give comparison of ATR-FTIR spectra of pure curcumin derivatives with composite materials.
  4. Give comparison of Fluorescence emission and UV-visible spectra of pure curcumin derivatives with composite materials.

Author Response

Dear reviewer,

Thank you for your appreciation of the study and kindly comments. The suggestions made by you will led to an increase in the scientific value of the paper.

Thus, we did the revision as reported below:

  1. The abstract and conclusion should precise and need to rewrite.

The abstract and conclusion have been improved:

„Abstract: Research in the field of natural dyes has constantly focused on methods of conditioning curcumin and diversifying their fields of use. In this study, hybrid materials were obtained from modified silica structures, as host matrices, in which curcumin dyes were embedded. The influence of the silica network structure on the optical properties and the antimicrobial activity of the hybrid materials were followed. By modifying the ratio between phenyltriethoxysilane:diphenyldimethoxysilane (PTES:DPDMES), it was possible to evaluate the influence of organosilane network modifiers on the morphostructural characteristics of nanocomposites. The nanosols obtained by the sol-gel method, in acid catalysis. The nanocomposites obtained, were deposited as films on glass support and showed a transmittance value (T measured at 550 nm) of around 90% and reflectance of about 11%, comparable to the properties of the uncovered support. For the coatings deposited on PET (polyethylene terephthalate) films, these properties remained on average of T550 = 85% and R550 = 11% without significantly modifying the optical properties of the support. Sequestration of the dye in silica networks reduced the antimicrobial activity of the nanocomposites obtained, compared to native dyes. Tests performed on Candida albicans fungi showed good results for the two curcumin derivatives embedded in silica networks (11-18 mm) by the spot inoculation method, compared to the spot diameter of 20-23 mm alcoholic dye solution. Also, hybrids with the CA derivative were the most effective (halo diameter of 17-18 mm) in inhibiting the growth of Gram-positive bacteria, compared to the curcumin derivative in alcoholic solution (halo diameter of 21 mm).  The results of the study showed that the presence of 20-40% by weight DPDMES in the composition of nanosols is the optimal range for obtaining hybrid films that host curcumin derivatives, with potential uses in the field of optical films or bioactive coatings.”

Conclusions       

„In this study we looked at the influence of the characteristics of the siloxane host matrix on the optical and antimicrobial properties of curcumin derivatives. In this sense, four types of host matrices were obtained by modifying the ratio of alkoxysilanes (PTES:DPDMES) used as network modifiers, in order to accommodate embedded chromophores. The nanocomposites obtained were deposited as films on glass support and showed a transmittance value of around 90% and reflectance of about 11%, similar to the values of the uncovered support. For the coatings deposited on PET films, these properties remained around 85% for T550 and of 11% for R550 without significantly modifying the initial optical properties of the support. Evaluations of the antimicrobial activity of nanocomposites were observed by the method of disc and spot inoculation. The test results reveal the existence of antimicrobial activity of nanocomposites by the formation of a halo with different diameters around the sample that depend on the architecture of the matrix in which the curcumin derivative is incorporated. After the morpho-structural characterization of nanocomposites and the evaluation of optical properties, resulted that the presence of 20-40% DPDMES in the composition of nanosols is the range in which it is possible to optimize organic-silica hybrid films hosting curcumin derivatives, useful for applications in the field of solar cells, food packaging or as bioactive coatings.”

  1. The authors used two curcumin derivatives to embed in siloxane polymer matrix, synthesized in laboratory. To increase more clarity about synthesized material, give comparison of decomposition of pure curcumin derivatives with composite materials.

It was completed with Figure 4 c, which illustrates the steps of decomposition of pure dyes.

Figure 4. Thermogravimetric curves of hybrid materials, with different siloxane host matrices (a) or different dyes (b) and pure dyes (c).

“This decomposition process is directly related to the degradation of the siloxane matrix, the pure dyes showing decomposition steps at lower temperatures (Figure 4c). As shown in Table 2, the maximum mass loss occurs at 262°C for curcumin and 296°C for the CA derivative.”

  1. Give comparison of ATR-FTIR spectra of pure curcumin derivatives with composite materials.

Figure 3 was completed with 3b containing the ATR-FTIR spectra of the dyes.

Figure 3. ATR-FTIR spectra of siloxane-type host matrices (a) and spectra of pure dyes and dyes embedded in matrices (b).

“Because the content of dyes in the film-forming material is very small, their structural features are more difficult to detect by FTIR spectrometry. However, in the case of CA there are two characteristic functional groups that have very high intensity, namely the acetamide group, which has the band characteristic of the carbonyl group located at 1662 cm-1 and the band characteristic of the keto-enol equilibrium of the acetylacetone rest, located at 1633 cm-1. These are found in the FTIR spectrum of M1-CA, embedded in the wide and asymmetric band, characteristic of water bending vibration, which has a maximum at 1623 cm-1, as it can be seen from the inset of Figure 3b.”

  1. Give comparison of Fluorescence emission and UV-visible spectra of pure curcumin derivatives with composite materials.

It was completed with a short comparison between the optical properties of pure dyes and curcumin derivatives embedded in siloxane matrices (line no. 310 and line no. 336):

“These maxima are given by the presence of the dye in the matrix, with a bathofluoric shift due to intermolecular interactions by hydrogen bonds. Pure curcumin derivatives in the polar environment of the silica matrix are characterized by fluorescent emission peak at 545 nm for curcumin and 530 nm for the CA derivative, respectively [63].”

“It can be seen that the properties of the coatings in which the two chromophores are present are similar along the visible spectrum, in the absorption range of pure dyes at 424 nm for CC and 419 nm for the CA derivative [63].”

I hope that most of the English mistakes and typos have been corrected.

Due to the recommendations of the other reviewers, the text was slightly modified.

Best regards,

Monica Raduly

Reviewer 2 Report

The authors were successfully able to develop siloxane host matrix on the optical and antimicrobial properties of curcumin derivatives. The effect of organosilane network modifiers on the morphostructural features of nanocomposites were studied by changing the ratio of PTES: DPDMES (phenyltriethoxysilane:diphenyl dimethoxysilane). After  morphostructural characterization of nanocomposites and the evaluation of optical properties, 20-40% DPDMES in the composition of nanosols was selected as the optimum range for obtaining of hybrid films that host curcumin derivatives, with potential uses in the field of solar cells, food packaging orbioactive coatings.  

  1. Please add a scheme in the introduction illustrating the synthesis of the siloxane polymer matrix based nanocomposites
  2. In the study of the antimicrobial activity of nanocomposites, the comparison of antimicrobial activity with standard antibiotics against Candida albicans, Staphylococcus aureus and Escherichia coli should also be done in order to validate the antimicrobial activity of the nanocomposites.
  3. Line No: 188

    What could be the major reason behind the biggest loss of mass in the case of M4-CC, since M4-CC is composed of PTES and DPEMES. Further, as explained DPDMES is affecting the weight loss, M2-CC has more amount than M4-CC.

Author Response

Dear reviewer,

Thank you for your appreciation of the study and for your kindly comments. The suggestions made by you will led to an increase in the scientific value of the paper.

Thus, we did the revision as reported below:

  1. Please add a scheme in the introduction illustrating the synthesis of the siloxane polymer matrix based nanocomposites

A scheme illustrating the synthesis of nanocomposites has been added.

  1. In the study of the antimicrobial activity of nanocomposites, the comparison of antimicrobial activity with standard antibiotics against Candida albicans, Staphylococcus aureus and Escherichia coli should also be done in order to validate the antimicrobial activity of the nanocomposites.

A brief comparison between nanocomposites and antibiotics (line no. 473) and pictures (figure 10) with the results of antimicrobial antibiotic tests was added.

“Although it does not exceed the level of antimicrobial activity of Ciprofloxacin (Figure 10), the existence of antibacterial properties of nanocomposites can be noticed. These properties can be improved by increasing the amount of dye in the nanosols composition. Regarding the antifungal properties, the results of the tests reveal an activity comparable to that of Fluconazole, an antibiotic known in the treatment of fungal infections [68,69].”

Figure 10. The antimicrobial activity of hybrid materials with (a) CC, (b) CA and (c) the difference between the antimicrobial activity of the alcoholic solutions of the native dyes, that of the dyes encapsulated in siloxane matrices and that of the antibiotics (*Fluconazole (25 µg/ml), **Ciprofloxacin (5 µg/ml))

  1. Line No: 188

What could be the major reason behind the biggest loss of mass in the case of M4-CC, since M4-CC is composed of PTES and DPEMES. Further, as explained DPDMES is affecting the weight loss, M2-CC has more amount than M4-CC.

It was supplemented with explanations on mass losses for M4-CC (line no 243).

„However, the highest mass loss occur in the case of M4-CC nanocomposites, whose composition lacks the network generator. The absence of TEOS in the nanosol affects the architecture of the film, due to the lack of highly crosslinked organized structures that are generated. For this reason, the intermolecular forces of p-p or van der Waals type present in M4-CC are easier to break and lead to higher mass loss. The absence of TEOS is also reflected in the reduced amount of residue, respectively 12% compared to the initial mass.”

I hope that most of the English mistakes and typos have been corrected.

Due to the recommendations of other reviewers, the text was slightly modified.

Best regards,

Monica Raduly

Reviewer 3 Report

The research should be addressed and the manuscript needs to be revised for a more clear explanation. For example,

Abstract

- The abstract is needed to modify and shows the requirement properties for optical application or bioactive coating.

- PTES:DPDMES (phenyltriethoxysilane:diphenyldimethoxysilane) … it should be “ phenyltriethoxysilane:diphenyldimethoxysilane (PTES:DPDMES)”

Introduction

- The background of varying of PTES:DPDMES did not mention in introduction.The review of background of this material need to be added.

- Introduction shows the lack of use curcumine for coating.

- The introduction should shows the relevant information to the topic, it need a revision.

Experiment

- “2.1.1. Curcumin dyes 69 Curcuminoid-like derivatives have been synthesized at microwaves and purified 70 using a method, already published by our group [51].”

The brief explanation should be added.

- All specific property of material has to address in 2.1.2. The property of polyethylenterephthalate (PET) was missing.

- 2.1.3. Microorganisms, the strains of bacteria should be addressed.

Author Response

Dear reviewer,

Thank you for your appreciation of the study and kindly comments. The suggestions made by you will led to an increase of the scientific value of the paper.

Thus, we did the revision of the paper, as reported below:

1) Abstract

- The abstract is needed to modify and shows the requirement properties for optical application or bioactive coating.

- PTES:DPDMES (phenyltriethoxysilane:diphenyldimethoxysilane) … it should be “ phenyltriethoxysilane:diphenyldimethoxysilane (PTES:DPDMES)”

 The abstract was modified and supplemented with the results obtained after testing the optical and antimicrobial properties:

“Research in the field of natural dyes has constantly focused on methods of conditioning curcumin and diversifying their fields of use. In this study, hybrid materials were obtained from modified silica structures, as host matrices, in which curcumin dyes were embedded. The influence of the silica network structure on the optical properties and the antimicrobial activity of the hybrid materials were followed. By modifying the ratio between phenyltriethoxysilane:diphenyldimethoxysilane (PTES:DPDMES ), it was possible to evaluate the influence of organosilane network modifiers on the morphostructural characteristics of nanocomposites.The nanosols obtained by the sol-gel method, in acid catalysis.. The nanocomposites obtained, were deposited as films on glass support and showed a transmittance value (T measured at 550 nm) of around 90% and reflectance of about 11%, comparable to the properties of the uncovered support. For the coatings deposited on PET (polyethylene terephthalate) films, these properties remained on average of T550 = 85% and R550 = 11% without significantly modifying the optical properties of the support. Sequestration of the dye in silica networks reduced the antimicrobial activity of the nanocomposites obtained, compared to native dyes. Tests performed on Candida albicans fungi showed good results for the two curcumin derivatives embedded in silica networks (11-18 mm) by the spot inoculation method, compared to the spot diameter of 20-23 mm alcoholic dye solution. Also, hybrids with the CA derivative were the most effective (halo diameter of 17-18 mm) in inhibiting the growth of Gram-positive bacteria, compared to the curcumin derivative in alcoholic solution (halo diameter of 21 mm).  The results of the study showed that the presence of 20-40% by weight DPDMES in the composition of nanosols is the optimal range for obtaining hybrid films that host curcumin derivatives, with potential uses in the field of optical films or bioactive coatings.”

2) Introduction

- The background of varying of PTES:DPDMES did not mention in introduction.The review of background of this material need to be added.

- Introduction shows the lack of use curcumine for coating.

- The introduction should show the relevant information to the topic, it need a revision.

The "Introduction" chapter has been supplemented with data from the literature on sol-gel coatings.

“At the same time, one of the most common methods of obtaining film-forming materials is the sol-gel process. By this method a series of nanocomposites, which are based on the silica network generated by triethoxysilane modified with different types of organic silane derivatives with alkyl or aromatic groups, can be made. They have the role of creating certain morphological structures into the network, which can modify the general properties of nanocomposites and / or can create environments compatible with other organic or inorganic structures, which are to be incorporated in the siloxane matrix. Thus, organic-inorganic hybrid materials are obtained and deposited as thin films [51-53]. Depending on the field in which they are used, the nanocomposites must have certain properties and be compatible with the support on which they are deposited [54,55]. In other cases, anticorrosive coatings must have barrier properties and usually contain metal oxides, photoactive coatings used in solar applications, must have good anti-reflection and transmitting properties, and usually contain dyes, or metal complexes, while anti-biofilm coatings, frequentely host metallic nanoparticles, natural compounds or antibiotics and have antimicrobial activity [56-62].”

3) Experimental

- “2.1.1. Curcumin dyes 69 Curcuminoid-like derivatives have been synthesized at microwaves and purified 70 using a method, already published by our group [51].”

The brief explanation should be added.

It was completed with a brief presentation of the method for the synthesis of curcumin derivatives.

“A mixture of boron trioxide (4 mmol), acetylacetone (8 mmol) and tributyl borate (3.2 mmol) was introduced in a porcelain capsule and was irradiated in a microwave oven at 300 W for 10 min. After the formation of the boron complex of acetylacetone, aromatic aldehyde (7 mmol) and dodecylamine (0.162 mmol) were added and then the mixture was irradiated for another 20 minutes at 100-500W (depending on the aromatic aldehyde chemical structure). To the reaction mixture was added an aqueous solution of acetic acid (10% by weight) and the obtained suspension was filtered off, and the solid product was washed with cold water and dried. The obtained product was purified by recrystallization from a mixture of ethyl-acetate : methanol = 3:2 (v/v) [63].”

- All specific property of material has to address in 2.1.2. The property of polyethylenterephthalate (PET) was missing.

Properties of the PET films were completed as follows:

“The obtained films were deposited on biaxially-oriented polyethylene terephthalate films (PET, Tg=105°C, 0.075 mm thick, Xerox Co., Norwalk, Connecticut, USA) and on microscope slides (ISOLAB, 1 mm thick, purchased from AMEX, Bucharest, Romania).”

- 2.1.3. Microorganisms, the strains of bacteria should be addressed.

It was completed with the strains  numbers:

“For the evaluation of antibacterial activity the following strains of bacteria were used: Staphylococcus aureus, ATCC 25923 (S.aureus), Escherichia coli, ATCC 25922 (E.coli) and strain of fungi: Candida albicans, ATCC 10231(C. albicans)”.

I hope that most of the English mistakes and typos have been corrected.

Due to the recommendations of the other reviewers, the text was slightly modified.

Best regards,

Monica Raduly

Round 2

Reviewer 2 Report

The paper need to be improved in terms of writing, lots of grammatical errors still persists. In addition, the authors need to address CA is meant to be contact angle or curcumin derivative?

Author Response

Dear reviewer,

Thank you for your appreciation of the study and kindly comments.

We gave up the abbreviation of the contact angle - CA, in favor of the dye. Thus, in the text, where CA was the abbreviation of the contact angle, it was deleted and replaced with "contact angle".

I hope that most of the mistakes in English and typing have been corrected.

Best regards,

Monica Raduly